# A novel immunofluorescent test system for SARS-CoV-2 detection in infected cells

**Alexandra Rak** [1]*, **Victoria Matyushenko**[1], **Polina Prokopenko**[1], **Arina Kostromitina**[1], **Dmitry Polyakov**[2], **Alexey Sokolov**[2], **Larisa Rudenko**[1], **Irina Isakova-Sivak**[1]

1 Department of Virology, Institute of Experimental Medicine, St. Petersburg, Russian Federation,
2 Department of Molecular Genetics, Institute of Experimental Medicine, St. Petersburg, Russian Federation

* rak.ay@iemspb.ru

**Data Availability Statement:** The minimal data cannot be shared publicly because of its confidentiality. The dataset used to prepare the article represents optical density and FFU values in

## Abstract

Highly variable pandemic coronavirus SARS-CoV-2, which causes the hazardous COVID-19 infection, has been persistent in the human population since late 2019. A prompt assessment of individual and herd immunity against the infection can be accomplished by using rapid tests to determine antiviral antibody levels. The microneutralization assay (MN) is one of the most widely used diagnostic methods that has been proposed to assess the qualitative and quantitative characteristics of virus-specific humoral immunity in COVID-19 convalescents or vaccine recipients. However, some aspects of the assay, such as sensitivity and time cost, need improvement. Here, we developed an express test, which may be potentially used in clinical practice for the assessment of serum-caused SARS-CoV-2 inhibition in infected cell cultures. It implies the detection and counting of coronaviral fluorescent-forming units (FFU) and includes two sequentially used developing components: biotinylated mouse monoclonal antibodies against the recombinant N protein of SARS-CoV-2 (B.1) and the recombinant EGFP-streptavidin fusion protein. Due to the universal specificity of the antibodies, our analytical tool is suitable for the detection of various strains of SARS-CoV-2 when determining both the infectious titer of viruses and the titer of serum virus-neutralizing antibodies. The developed two-component test system is characterized by high sensitivity, a reduced number of analytic stages and low assay cost, as well as by flexibility, since it may be modified for detection of other pathogens using the appropriate antibodies.

## 1 Introduction

COVID-19 is an acute respiratory infection caused by the coronavirus SARS-CoV-2, characterized by high contagiousness and mortality of the infected population, which determines its high socio-economic significance. The global COVID-19 pandemic, which began in late 2019, has affected more than 704 million people and took the lives of 7 million victims [1]. Despite to date the pandemic is finished, the disease cases continue to emerge in the form of successive waves of the worldwide spread of new SARS-CoV-2variants, which is associated with the high antigenic variability of the virus [2].

the plate wells, so uploading them to public repositories without descriptions of groups, samples and dilutions (including location of samples within the plates) seems to be inappropriate. We uploaded a raw dataset related to our manuscript to the repository of the Local Ethics Committee of the Institute of Experimental Medicine (email: iemetcom@yandex.ru), located on Yandex Disk at: https://disk.yandex.ru/client/disk/Datasets%20for%20External%20Requests/Rak%20et%20al.%202024%20PLoS%20One%20(FFU%20test%20system)%20raw%20data%20set Access to this repository is provided by the Local Ethics Committee staff using the login iemetcom@yandex.ru, and data may be stored there indefinitely. The minimal data are available for researchers who meet the criteria for access to confidential data upon request to the following member of the IEM Local Ethics Committee who is responsible for ensuring data access: Dr. Olga V. Kirik, email: olga_kirik@mail.ru, mob: +7 (951) 654-94-23.

**Funding:** This research was funded by the RSCF grant 21-75-30003. The funder had no role in study design, data collection and analysis, decision to publish, or preparation of the manuscript.

**Competing interests:** The authors have declared that no competing interests exist.

The rapid and precise COVID-19 diagnostics are required to control the spread of infection and to assess the population immunity. There are several widely used methods of express COVID-19 testing, such as PCR, the detection of viral antigens, or serological measurement of antiviral antibody titer by ELISA, LFIA, CLIA or microneutralization assay (MN) assay [3]. Despite the fact that MN reaction, as one of the serological approaches, is not applicable to reveal the acute phase of COVID-19, it still considered to be "the gold standard" to assess both the intensity of the formation of antiviral antibodies and their functional activity, that is, the ability to abolish the viral propagation [4]. These data are important not only for individual COVID-19 diagnostics, but also for assessment of the rate of infection spread and herd immunity state [5].

The majority of modern serological test systems for SARS-CoV-2 detection involve the use of the surface Spike protein as a target antigen for virus detection, which seems appropriate since it is used to penetrate target cells [6]. However, it does not allow determining the cause of the antiviral antibodies appearance (as a result of vaccination or infection) and is not suitable for detection of recent SARS-CoV-2 variants that differ mainly in this antigen [7]. In this regard, it seems feasible to develop analytical test systems based on more conservative SARS-CoV-2 antigens, in particular using the highly immunogenic nucleocapsid protein (N) actively produced by infected cells [8].

Beyond packaging and compartmentalization, the viral genome, the N protein of SARS-CoV-2 also serves as the transcription enhancer for viral mRNAs and as a suppressor of interferon-mediated defense reactions [9]. Importantly, this protein was detected both in the cytoplasm and at the cell surface [10]. This makes N protein a promising target for designing antibody- or antigen-based diagnostic test systems as well as for the development of broadly reactive COVID-19 vaccines inducing the generation of antibodies that mediate the killing reactions.

It is widely accepted that N protein is highly conserved, and that N-based vaccines and diagnostics should not be regularly updated. However, the N protein is also slowly being modified by viral evolution. It is known that some mutations may change the N protein antigenicity [11], so there is a possibility that diagnostic antibodies will not recognize the mutated viruses. It means that anti-N monoclonal antibodies used in these assays should have universal specificity, meaning that they should be suitable for detection of all SARS-CoV-2 variants.

Here, we used one of the previously generated anti-N (B.1) mAbs to develop an in-house test system for universal detection of focus-forming units (FFU) in virus-infected cells. We compared the infectious titers calculated as FFU per mL and as $TCID_{50}$, and tested several serum samples from COVID-19 patients using our test system and previously reported assay. Our results suggest that a novel test system may be used for the *in vitro* SARS-CoV-2 detection, in particular for MN assay development.

## 2 Materials and methods

### 2.1. Cells, viruses and monoclonal antibodies

The following SARS-CoV-2 viruses belonging to different lineages were obtained from the Smorodintsev Research Institute of Influenza (Saint Petersburg, Russia):

- hCoV-19/St_Petersburg-3524S/2020 (B.1 Lineage, Wuhan)

- hCoV-19/Japan/QK002/2020 (B.1.1.7 Lineage, Alpha)

- hCoV-19/St_Petersburg-27029/2021 (B.1.371 Lineage, Beta)

- hCoV-19/Japan/TY7/503-2021 (P.1 Lineage, Gamma)

- hCoV-19/Russia/SPE-RII-32759V/2021 (B.1.617.2 Lineage, Delta)

- hCoV-19/Russia/SPE-RII-6086V1/2021 (B.1.529.1 Lineage, Omicron)

African green monkey kidney Vero E6 cells were obtained from the American Type Culture Collection (ATCC) and routinely maintained in DMEM supplemented with 10% fetal bovine serum (FBS) and 1× antibiotic–antimycotic (AA) (all from Capricorn Scientific, Ebsdorfergrund, Germany).

The viruses were propagated on the Vero E6 cells using DMEM supplemented with 2% FBS, 10 mM of HEPES and 1× antibiotic-antimycotic (all from Capricorn Scientific, Ebsdorfergrund, Germany) at 37°C and 5% $CO_2$. All experiments with live SARS-CoV-2 were performed in a biosafety-level-3 laboratory (BSL-3).

NCL5 antibodies against the N protein of SARS-CoV-2 (B.1), specifically binding N proteins of B.1.351, P.1, B.1.617.2 and B.1.1.529 VOCs were previously obtained [11] using standard hybridoma approach, isotyped as IgG2a, dialyzed against 20 mM phosphate buffer, pH 7.4 (PBS), aliquoted and stored at -20°C.

## 2.2. Antibody labeling

For biotinylation of antibodies, a 4.4 mM biotin-X-activated ester (Lumiprobe, Russia) in DMSO was added to the purified monoclonal antibodies NCL5 and incubated for 18 hours at +4°C. Next, the reaction was stopped by adding 5% of 4.4 mM lysine-HCl in PBS, and the resulting biotin-conjugated antibody was dialyzed against PBS.

## 2.3. Study participants

One hundred and one serum samples were collected during the period from 2 December 2020 to 12 December 2023 from fourty subjects fully vaccinated with Sputnik V and fifty-nine COVID-19 convalescents aged 31 to 84 years, who participated in the study of humoral and T-cell responses to SARS-CoV-2. The research was approved by the Ethics Committee of the Institute of Experimental Medicine (protocol No. 2/20 on 7 April 2020). The disease onset ranged between May 2020 and September 2022, and time post symptoms onset (PSO) was from 1 to 16 months. For Sputnik V-vaccinated individuals, serum samples were collected 1 to 6 months after the second vaccine dose. A written informed consent was obtained from all participants. Control historical serum samples (n = 5) were obtained from archived specimens collected on 7–24 November 2012 from healthy adults who participated in Phase I trials of H5N2 live attenuated influenza vaccine (LAIV) (NCT01719783) [12].

## 2.4. EGFP-streptavidin expression and purification

Genes encoding EGFP and streptavidin (SA) were cloned into multiple cloning site (MCS) of the pET22b (+) expression vector using *NdeI*, *BamHI* and *HindIII* restriction sites within a common reading frame. In this plasmid, the gene inserted in MCS yielded a recombinant protein with a poly-histidine tag at the C-terminus. The recombinant protein then was expressed in *E. coli* BL21 (DE3) cells using a standard protocol [13].

Purification of EGFP-SA was performed using the immobilized metal affinity chromatography (IMAC) on a column packed with His-Bind resin (Novagen, USA) charged with $Co^{2+}$ [14]. The flow rate at all fractionation stages was 0.5 mL/min. To remove non-specifically bound proteins, wash buffer (20 mM Tris-HCl, 0.5 M NaCl, 60 mM imidazole, pH 7.9) was used. The bound recombinant EGFP-SA was eluted with buffer containing 300 mM NaCl and 250 mM imidazole. The protein detection was performed at a wavelength of 280 nm and 488 nm using Implen™ NanoPhotometer™ N60.

SDS-PAGE of collected fractions was performed using a running gel with 10% acrylamide concentration and stacking gel containing 5% acrylamide. Protein samples (0.3–0.5 mg/mL) were dissolved in loading buffer and denatured for 5 min at 95°C, then 15 μL of each was loaded onto the gel (1–1.5 μg of protein per lane). SDS-PAGE was performed in Mini-PRO-TEAN Tetra Cell system (BioRad Hercules, CA, USA) and the results were visualized by standard staining with Coomassie Brilliant Blue G-250.

Native PAGE was performed by previously reported protocol [15] in 7.5% running gel and 5% stacking gel prepared on 375 Tris-HCl buffer, 8.9. Samples containing 2 μg of protein were mixed with 62,5 Tris-HCl loading buffer, 6.8 supplemented with 0,001% bromophenolic blue and 50% glycerol and run at +4° and 20 V/sm in Mini-PROTEAN Tetra Cell system (BioRad Hercules, CA, USA) with following detection by Coomassie staining or immunoblotting.

Western blot analysis of EGFP-SA was performed by wet protein transfer to 0.45 μm nitrocellulose membrane using the Mini-PROTEAN Tetra Cell chamber, followed by the blockage of non-specific binding with 5% skim milk in PBS+0.05% Tween-20 (PBS-T). Then, the membrane was treated with the biotynilated HRP (Thermo Fisher Scientific, Waltham, MA, USA), and developed by 0.05% diaminobenzidine (DAB) in PBS containing 1% hydrogen peroxide.

## 2.5. SARS-CoV-2 titration

Viral titer was determined by previously described 50% Tissue Culture Infection Dose ($TCID_{50}$) assay [16]. In brief, 10-fold dilutions of viruses in DMEM/2%FBS were added to a 96-well cell culture plates 95%-monolayered with Vero E6 cells and incubated for 72 h at 37°C and 5% CO2. The appearance of cytopathic effect (CPE) was used as an indicator of cell infection, and then the plates were fixed with 10% formaldehyde in PBS for 18 h at 4°C. After fixative removing and triple washing with PBS-T, the cells were permeabilized with 0.1% Triton-X100 for 15 min, treated with 3% hydrogen peroxide for 15 min at 37°C and then blocked with 5% skim milk in PBS at 37°C for 1 h. Next, NCL5 antibodies (2 μg/mL in PBS-T) were added for 1 h and after washing, followed by addition of 1:3000 solution of goat-anti-mouse HRP conjugate (BioRad, USA). The plates were finally developed with 1-Step™ Ultra TMB-E-LISA Substrate Solution (Thermo Fisher Scientific, USA) and OD was read on xMark Microplate Absorbance Spectrophotometer (Bio-Rad). An infection titer was calculated using Reed and Muench method and expressed as $log_{10}TCID_{50}/mL$ [17].

Alternatively, viral titer was assessed by the fluorescent focus units (FFU) counting. For this, a day before the study, cells were seeded in DMEM medium (Capricorn, Germany) supplemented with 10% fetal bovine serum (FBS) in 96-well plates at a dose of $4 \times 10^5$ cells/well. Before inoculation, the monolayer was washed twice with PBS and then 10-fold dilutions of viruses were added in DMEM containing 2% PBS, 10 mM HEPES and 1× AA. Virus adsorption was carried out for 1 h, after which 100 μl of growth medium was added to the wells and the plates were incubated for 11–19 h at 37°C and 5% CO2. After the incubation, the growth medium was carefully removed, and the cells were fixed with 10% formaldehyde in PBS for 18 h at 4°C. Similarly, with the standard protocol, at the first analytical stage, the fixing solution was removed, the cells were washed three times with PBS-T and permeabilized with 0.1% Triton X-100 in PBS. Then, the plates were washed twice with PBS-T and treated with blocking buffer (5% skim milk in PBS-T) for 1 h at 37°C. At the next step, a solution of biotinylated NCL5 antibodies (2.5 μg/mL in PBS-T supplemented with 1% skim milk) was added to the wells and incubated at 37°C for 1 h. After washing three times, 4 μg/ml EGFP-SA in PBS-T was added and incubated at 37°C for 40 min. After a final triple washing, the plates were dried and analyzed using AID vSpot Spectrum (AID Autoimmun Diagnostika GmbH, Germany) in

fluorescent signal registration mode. The amount of FFU was counted in wells with 100–200 spots, and the viral titer was calculated by multiplying by the dilution factor and expressed as $\log_{10}$FFU/mL.

## 2.6. MN assay

The 50% virus-neutralizing titers were determined as previously described in [16] with minor modifications. Briefly, serum samples were incubated at 56˚C for 30 min to inactivate the complement system and then two-fold (from 1:10 to 1:640) diluted in DMEM + 2% FBS. Each serum sample was analyzed in duplicates. The serum dilutions were mixed with 300 $TCID_{50}$ of SARS-CoV-2 and incubated at 37˚C for 1 h. Then, the mixtures were transferred to Vero cells seeded on 96-well cell culture plates at a density of $4 \times 10^5$ cells/well one day prior to the analysis, whereby 300 $TCID_{50}$ of virus was added to the wells corresponding to the positive controls, and DMEM + 2% FBS was added to the negative control wells. The plates were incubated for 30 min at 37˚C and 5% $CO_2$, then the supernatants were removed and replaced by the corresponding initial serum dilutions (except for the control wells), and fresh DMEM supplemented with 2% FBS was added to each well to incubate the plates at 37˚C and 5% $CO_2$ for 48 h (in case of developing by cell ELISA) or 19 h (if FFU detection was planned). After incubation, the virus in plates was inactivated by 10% formaldehyde at 4˚C for 24 h. Then the fixed plates were transferred to BSL-2 laboratory to perform the cell ELISA or FFU counting by similar protocols for SARS-CoV-2 titer assessment described above. To calculate the $MN_{50}$ titer, four parameter non-linear regression analysis was used, based on the $OD_{450}$ values or FFU quantities in virus-containing and non-infected wells as described in [16].

## 2.7. Statistical analysis

Data were analyzed with the statistical tool of Graph Pad Prism 6 Software GraphPad Software, San Diego, CA, USA. A four-parameter logistical analysis was performed to calculate the MN assay results (IC50). The regression was based using two replicates per dilution, and the titer was calculated from the regression curve. The relationships between the measurements were evaluated by Spearman correlation test. The significance of differences between paired data was evaluated by Wilcoxon test ($p < 0.05$).

# 3 Results

## 3.1. EGFP-SA obtaining

The recombinant fusion protein EGFP-SA was expressed in IPTG-induced transformants of *E.coli* strain BL21(DE3) and purified from the biomass lysate using IMAC. The model of EGFP-SA based on the known X-ray structures of the components of the fusion protein is shown in Fig 1A. SDS-PAGE analysis confirmed the predicted molecular mass of EGFP-SA (about 42 kDa) as well as purity of the obtained protein (Fig 1C). Furthermore, assessment of fusion protein stability at different storage conditions revealed it instability after the thaw-freezing cycles, suggesting that functionality of the protein can only be maintained when stored at +4˚C (Fig 1D–1F).

## 3.2. Assessment of infectious titers of different SARS-CoV-2 VOCs

We first compared the performance of the FFU detection assay for determining the infectious titer of SARS-CoV-2 compared to the standard $TCID_{50}$ assay. Importantly, we were only able to detect fluorescent focuses when the NCL5 antibody was biotinylated followed by the addition of the new EGFP-SA protein, but not when the antibody was conjugated to FITC (Fig 2).

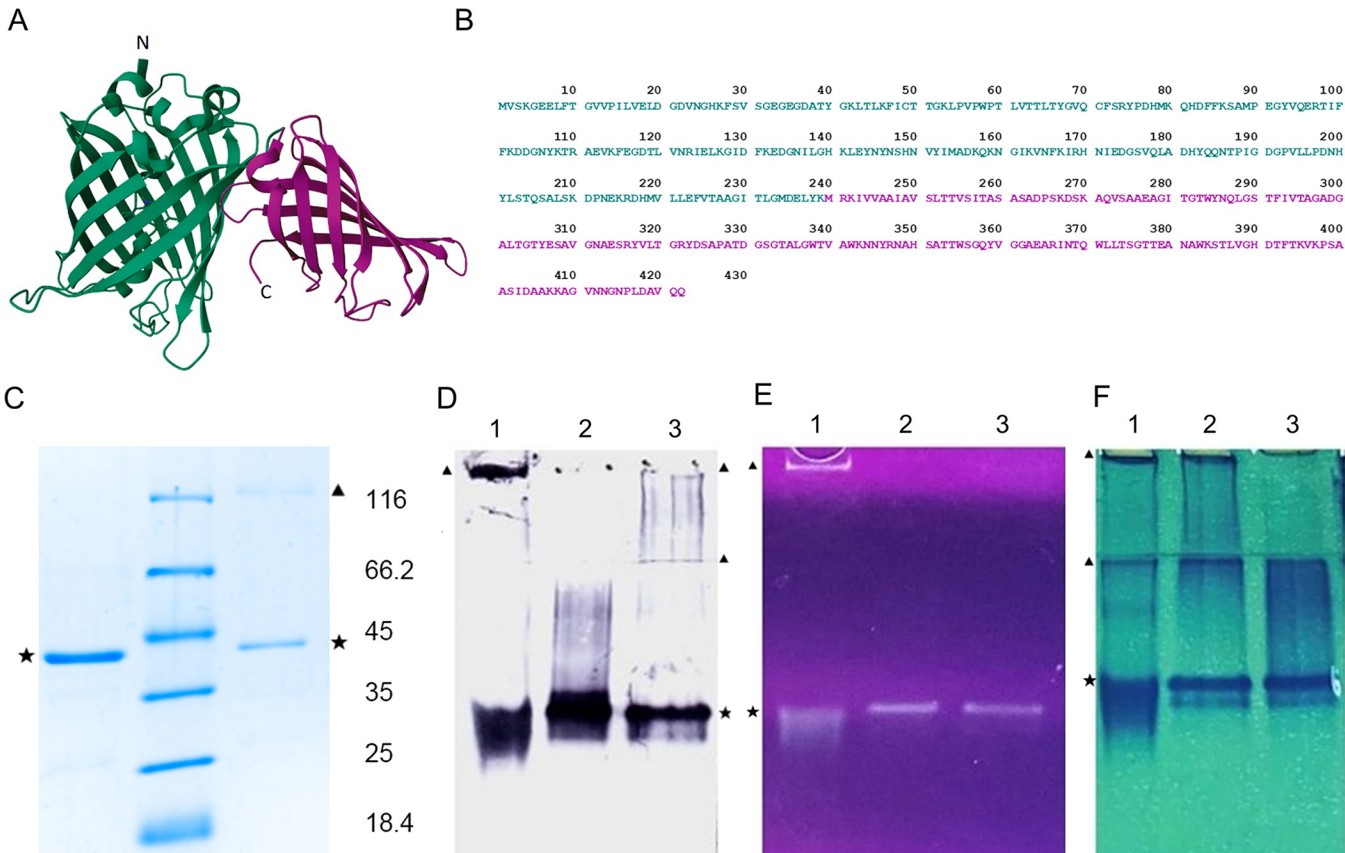

**Fig 1. Characterization of purified EGFP-SA.** (A) A model of EGFP-SA molecule based on the known X-ray structures of EGFP (PDB ID 2Y0G, shown in green) and monomeric streptavidin (PDB ID 6ZYT, shown in magenta); (B) amino acid sequence of EGFP-SA based on the sequencing results; (C) SDS-PAGE of the purified EGFP-SA, Coomassie-stained Native PAGE of the purified EGFP-SA followed by: (D) Western blot developed by a horseradish peroxidase conjugate with biotin; (E) detection in ultraviolet light; (F) the standard Coomassie G-250 staining. Lanes: +,—–SDS-PAGE under reducing or non-reducing conditions (with or without 2% beta-mercaptoethanol), respectively; 1 –fusion protein stored for 2 weeks at -20˚C with the addition of 50% glycerol; 2 –fusion protein stored for 2 weeks at +4˚C with the addition of 50% glycerol; 3 –freshly obtained fusion protein; ▲–aggregates of protein formed mostly under inadequate storage conditions; ★–bands corresponding to the active form of EGFP-SA.

It should be noted, that different SARS-CoV-2 variants grow with different kinetics on Vero E6 cells, and clearly defined focuses appear at different incubation time. For example, the B.1 ancestral strain produced focuses as early as 11 h post infection, whereas Omicron variant requires at least 19 hours of incubation to develop FFUs. Therefore, for each particular SARS-CoV-2 strain a preliminary kinetics experiment was performed to find optimal incubation time for FFU assay. Overall, using the biotin/ EGFP-SA detection system, virus titers of different SARS-CoV-2 variants were readily determined by FFU counting, and these results were in good agreement with virus titers determined by cellular ELISA as $TCID_{50}$ (Table 1). These data suggest that the new immunofluorescent test system can be used for determining SARS-CoV-2 neutralizing antibody titers in serum samples of individuals recovered from COVID-19 or vaccinated with COVID-19 vaccines.

### 3.3. Assessment of MN titers in COVID-19 convalescents and Sputnik V vaccinated individuals

To demonstrate that the newly developed immunofluorescent test system can be used for determining SARS-CoV-2 neutralizing antibody titers in human sera, we performed a side-by-

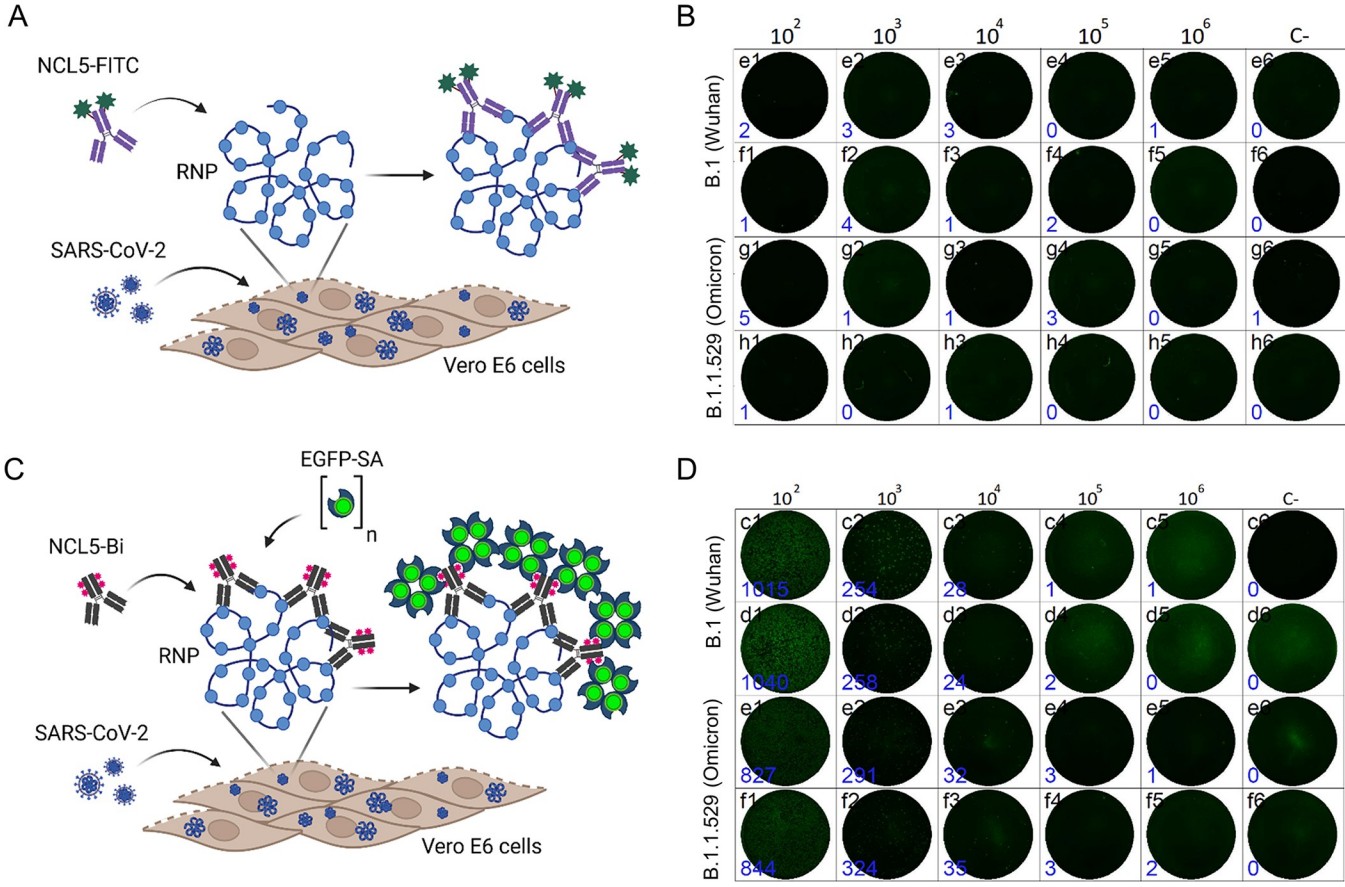

**Fig 2. Detection of SARS-CoV-2 in infected cells by fluorescent focus units (FFU) assay.** The scheme of FFU detection assay using FITC-labeled NCL5 antibody (A) or biotinylated NCL5 antibody and EGFP-SA (C). FFUs formed by SARS-CoV-2 of B.1 (Wuhan) and B.1.1.529 (Omicron) lineages were not detected in Vero E6 cells after 11 h and 16 h post-infection by FITC-labeled NCL5 antibody (B), but could be visualized and counted in case of using NCL5-Bi/EGFP-SA tandem (D).

side $MN_{50}$ assay of serum samples from COVID-19 convalescents and Sputnik V recipients using an in-house FFU test system and standard cell ELISA protocol [16] for detection of viral antigens in infected Vero E6 cells. Of note, FFU detection allowed data reading already on the next day after the $MN_{50}$ reaction setup, whereas standard protocol required at least three days until data acquisition. An example of the FFU data reading is shown in Fig 3A. Importantly,

**Table 1. Infection titers of different SARS-CoV-2 variants calculated using two independent methods.**

| Strain | Viral titer by the indicated assay, $\log_{10}$/mL | |
|---|---|---|
| | **FFU** | **TCID$_{50}$** |
| hCoV-19/St_Petersburg-3524S/2020 (Wuhan) | 5,8±0,2 | 6,4±0,3 |
| hCoV-19/Japan/QK002/2020 (Alpha) | 5,6±0,1 | 6,7±0,4 |
| hCoV-19/St_Petersburg-27029/2021 (Beta) | 5,6±0,1 | 6,7±0,3 |
| hCoV-19/Japan/TY7/503-2021 (Gamma) | 5,8±0,1 | 6,5±0,4 |
| hCoV-19/Russia/SPE-RII-32759V/2021 (Delta) | 6,5±0,3 | 7,0±0,3 |
| hCoV-19/Russia/SPE-RII-6086V1/2021 (Omicron) | 5,7±0,1 | 6,1±0,2 |

SD values are indicated as errors.

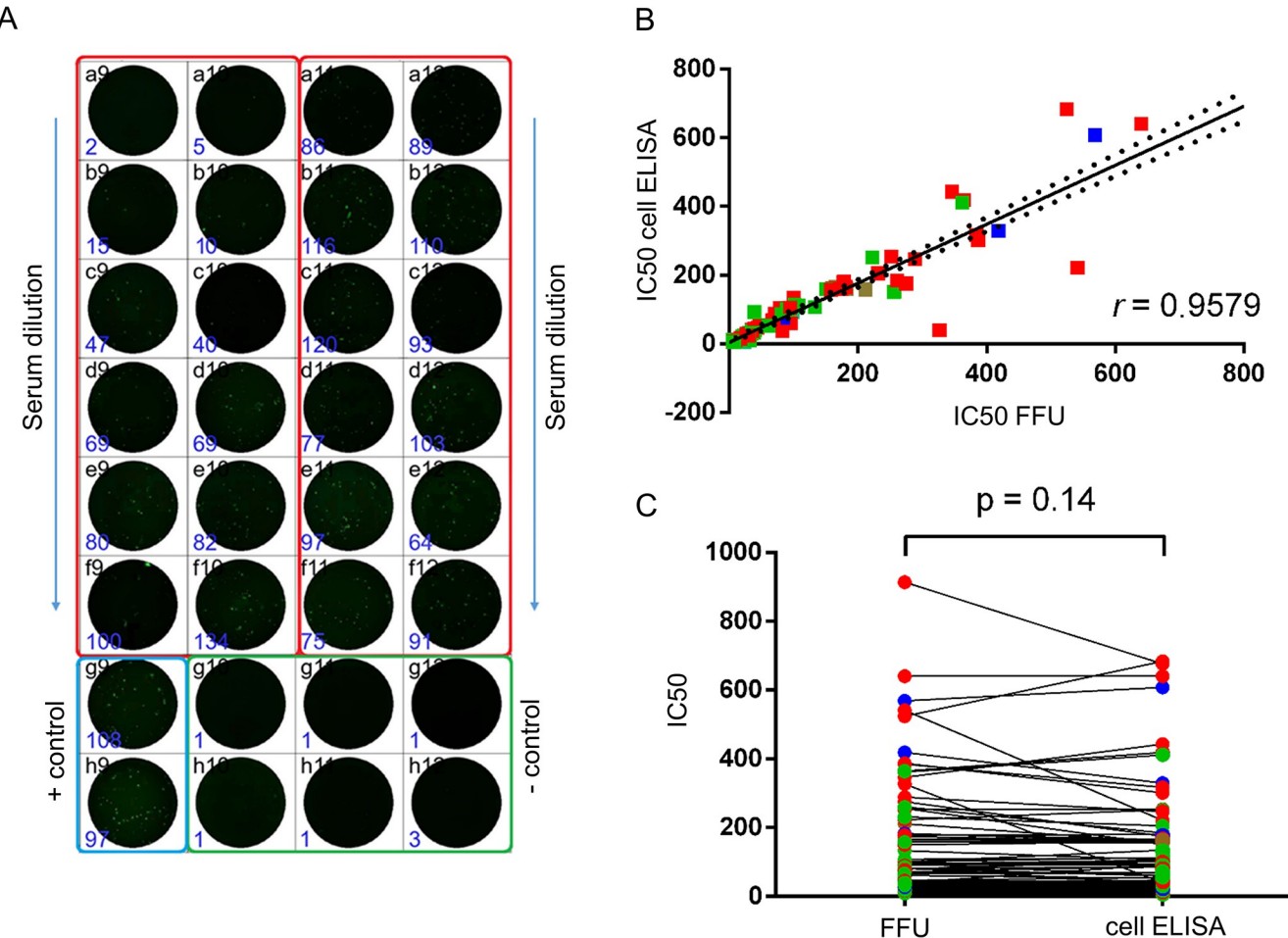

**Fig 3. Comparative evaluation of MN_50 titers (expressed as IC_50) of sera from vaccinated individuals (green) and B.1 (red), B.1.617.2 (blue), B.1.1.529 (brown) COVID-19 convalescents in infected Vero E6 cells by two methods.** (A) Appearance of FFU detection in MN assay; (B) correlation of $IC_{50}$ values obtained by FFU and $TCID_{50}$ assays; (C) Wilcoxon test for paired data revealed non-significant differences between $IC_{50}$ evaluation using FFU and $TCID_{50}$ assays.

the final MN50 values calculated using the FFU assay were in good correlation with the values obtained by the standard cell ELISA protocol (Fig 3B). Overall, there was no significant differences between the absolute $MN_{50}$ values determined by the two detection methods (Fig 3C).

## 4 Discussion

The neutralizing antibodies are detected in approximately 40%-70% of SARS-CoV-2-infected individuals, correlate with COVID-19 severity and are reported as key players in preventing viral entry into the host cell, so MN titer may be considered as an indicator of antiviral protection [18–20]. It is well known that serum IgG antibody levels, similarly with the serum neutralizing activity gradually decrease in COVID-19 convalescents, but don't disappear completely for a long time [21,22]. The maintenance of virus-neutralizing antibodies after SARS-CoV-2 infection was observed for up to several months after infection [18] and they may be boosted by revaccination. Thus, although MN titer measurement is not an appropriate method for acute COVID-19 testing, it seems to be suitable for evaluation of the humoral immunity persistence, especially in a context of possible revaccination requirements.

Several protocols have previously been proposed for performing MN assays to test SARS-CoV-2 inhibition by purified antibodies or serum [4,23–28]. The most widely used approaches to detect the virus in infected cells are non-specific assessment of virus titer by CPE and ELISAs implying antiviral monoclonal antibodies. Interestingly, Bennett et al. used the secondary antibody labeled with Alexa594 to detect SARS-CoV-2 and automatically count the percentage of infected cells [29]. In order to develop the test system with universal strain specificity, we used cross-strain N-specific biotinylated antibody and fluorescent developing fusion protein for the first time and reported the preliminary data on serum neutralizing activity in SARS-CoV-2 naïve donors, Sputnik V-vaccinated individuals and COVID-19 convalescents up to 16 months post-symptom onset, while the longevity of these responses will be further studied.

The results of MN titer assessment by developed approach and previously proposed method [16] were in good agreement. Our data suggest that in-house test has a better sensitivity due to the targeted FFU detection in virus-infected cells and signal amplification resulted from EGFP-SA binding to biotin-labeled antibody. Moreover, our test system has a number of additional benefits, such as a rapidness, cost-effectiveness, simplicity and flexibility of assay, because it may be modified for detection of other pathogens using the appropriate antibodies. As shown in the current study, the low doses of SARS-CoV-2 incubated for 19 h were sufficient to assess the virus-neutralizing activity of sera. This property is in good agreement with well-known fact that SARS-CoV-2 produces up to $10^3$ virions every 10 h with a potential to infect ACE2-positive cells [30]. In addition, FFU detecting test-system doesn't include any secondary anti-IgG conjugates, which is important feature for testing of SARS-CoV-2 inhibiting antibodies derived from model animals.

Our study is mainly limited by the absence of individual kinetics of serum neutralizing activity for each participant. However, we clearly demonstrated a direct correlation between the degree of serum dilution and a decrease in its virus-neutralizing activity using two independent approaches. Another limitation of the work is that the MN results obtained using the developed test-system were compared with only one contemporary method of SARS-CoV-2 detection in infected cells. Undoubtedly, additional studies are required to reveal the correlation between the titers of virus-neutralizing antibodies determined by a combination of biotinylated anti-N mAbs/EGFP-SA, differently labeled anti-N mAbs and Spike-targeting tests. An additional limitation of this study is the small number of analyzed serum samples; however, despite the low sample size, statistically significant correlation was found between the neutralizing titers assessed in two independent assays (Fig 3), suggesting that similar results will be obtained on a larger number of specimens in further studies. Despite we did not have access to sufficient serum samples from individuals immunized with mRNA vaccines due to the very limited availability of imported COVID-19 vaccines in our country, the results of the evaluation of sera neutralizing activity from people vaccinated with Sputnik V—the vector vaccine which also encoded only Spike protein, indicate that a strong correlation of MN results obtained by standard test and by our test system could be expected for samples from mRNA-vaccinated persons as well. Finally, an instability of EGFP-SA (Fig 1) should also be taken into account as a factor of test deficiency. The possibility of EGFP-SA stabilization without the affecting of its properties by commonly used protease inhibitors such as aprotinin, PMSF, benzamidine hydrochloride etc. and optimization of storage conditions (optimal temperature, lyophilization, glycerin addition etc.) should be studied in future.

## 5 Conclusion

The proposed assay of SARS-CoV-2 detection in infected cell cultures may be successfully used to rapidly evaluate the serum MN titers in patients who have recovered from COVID-19.

The described method seems to be a powerful tool to assess the levels of virus-neutralizing antibodies generated upon natural infection or administration of novel COVID-19 vaccine candidates. In contrast to traditional Spike-targeting tests designed to evaluate the humoral immune responses induced by infection or vaccination, our approach allows to estimate the capacity of vaccines to establish the cross-strain B-cell responses, which indicates its protective potential. Notably, this study used evolutionarily distant SARS-CoV-2 strains obtained upon different waves of COVID-19; therefore, given the advantages obtained, the developed FFU-targeting test system can be recommended for universal assessment of individual viral neutralizing antibody titers regardless of the SARS-CoV-2 strain that caused COVID-19.

## Supporting information

**S1 File. The blot/gel image data.** The file includes the original uncropped and unadjusted images underlying all blot or gel results reported in the figures.
(PDF)

## Acknowledgments

We thank Dr. Ekaterina Stepanova for her advices on SARS-CoV-2 propagation and technical assistance, as well as all blood donors who participated in the study.

## Author Contributions

**Conceptualization:** Alexandra Rak, Dmitry Polyakov, Irina Isakova-Sivak.

**Data curation:** Alexandra Rak, Victoria Matyushenko, Polina Prokopenko.

**Formal analysis:** Alexandra Rak, Victoria Matyushenko, Arina Kostromitina.

**Funding acquisition:** Irina Isakova-Sivak.

**Investigation:** Alexandra Rak, Victoria Matyushenko, Polina Prokopenko, Arina Kostromitina.

**Methodology:** Alexandra Rak, Victoria Matyushenko, Larisa Rudenko, Irina Isakova-Sivak.

**Project administration:** Larisa Rudenko, Irina Isakova-Sivak.

**Resources:** Dmitry Polyakov, Alexey Sokolov.

**Supervision:** Alexey Sokolov, Larisa Rudenko, Irina Isakova-Sivak.

**Visualization:** Alexandra Rak.

**Writing – original draft:** Alexandra Rak, Dmitry Polyakov.

**Writing – review & editing:** Victoria Matyushenko, Polina Prokopenko, Arina Kostromitina, Alexey Sokolov, Larisa Rudenko, Irina Isakova-Sivak.

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
