## [Decision Letter · Decision Letter 0]

27 Mar 2024

PONE-D-24-07586A novel immunofluorescent test system for SARS-CoV-2 detection in infected cellsPLOS ONE

Dear Dr. Rak,

Thank you for submitting your manuscript to PLOS ONE. After careful consideration, we feel that it has merit but does not fully meet PLOS ONE’s publication criteria as it currently stands. Therefore, we invite you to submit a revised version of the manuscript that addresses the points raised during the review process.

As suggested by the reviewer, you need to improve the figures and to extend the number of participants in the study.

We look forward to receiving your revised manuscript.

Kind regards,

Boyan Grigorov

Academic Editor

PLOS ONE

https://pdfs.semanticscholar.org/755b/684fa2e81cfa38957ed82f7fb90ee6d88405.pdf

https://www.mdpi.com/2076-393X/11/12/1810

In your revision ensure you cite all your sources (including your own works), and quote or rephrase any duplicated text outside the methods section. Further consideration is dependent on these concerns being addressed.

“This research was funded by the RSCF grant 21-75-30003.”

7.Please review your reference list to ensure that it is complete and correct. If you have cited papers that have been retracted, please include the rationale for doing so in the manuscript text, or remove these references and replace them with relevant current references. Any changes to the reference list should be mentioned in the rebuttal letter that accompanies your revised manuscript. If you need to cite a retracted article, indicate the article’s retracted status in the References list and also include a citation and full reference for the retraction notice.

Reviewers' comments:

Reviewer's Responses to Questions

**Comments to the Author**

1. Is the manuscript technically sound, and do the data support the conclusions?

Reviewer #1: Yes

2. Has the statistical analysis been performed appropriately and rigorously? 

Reviewer #1: Yes

3. Have the authors made all data underlying the findings in their manuscript fully available?

Reviewer #1: Yes

4. Is the manuscript presented in an intelligible fashion and written in standard English?

Reviewer #1: Yes

5. Review Comments to the Author

Reviewer #1: This publication proposes the development of a new analytical tool to detect SARS-CoV-2 by a two-component test system with high sensitivity, a reduced number of analytical steps and low cost.

The manuscript is well written and understandable.

# minor recomendations:

The population of vaccinated candidates could be extended to people vaccinated with other vaccines as Astra Zeneca, BioNtech or Moderna.

L100: "Thirty serum samples were collected during the period from 2 December 2020 to 12 December100

2023 from five subjects fully vaccinated with Sputnik V and twenty-three COVID-19 convalescents101

aged 31 to 84 years, who participated in the study of humoral and T-cell responses to SARS-CoV-2." For statistical reasons, the number of patients should be greater to validate the study.

Figure 1 could be improved with particular annotations on Westerblot (figure1C,D,E) for ease of reading.

Figure2 : the letters and numbers are not sufficiently legible on the tables (modification of size of letters for example). An explanation on negative and positive controls is to be added (in the text or on figures).

6. PLOS authors have the option to publish the peer review history of their article (what does this mean?). If published, this will include your full peer review and any attached files.

Reviewer #1: No

---

## [Author Response · Author response to Decision Letter 0]

9 May 2024

Thank you for consideration of our manuscript entitled "A novel immunofluorescent test system for SARS-CoV-2 detection in infected cells”. We appreciate the time and effort dedicated by the reviewer to provide insightful feedback on ways to strengthen our paper. Thus, it is with great pleasure that we resubmit for further consideration our article containing the changes that reflect the suggestions provided. We hope that our edits and responses which are provided below satisfactorily address all the issues and concerns the reviewer has noted.

To facilitate your review of our revisions, the following is a point-by-point response to the referee’s questions and comments.

Comments from Referee:

Reviewer #1: This publication proposes the development of a new analytical tool to detect SARS-CoV-2 by a two-component test system with high sensitivity, a reduced number of analytical steps and low cost.

The manuscript is well written and understandable.

Authors’ response: Thank you very much for a such positive feedback.

# minor recomendations:

The population of vaccinated candidates could be extended to people vaccinated with other vaccines as Astra Zeneca, BioNtech or Moderna. L100: "Thirty serum samples were collected during the period from 2 December 2020 to 12 December 2023 from five subjects fully vaccinated with Sputnik V and twenty-three COVID-19 convalescents aged 31 to 84 years, who participated in the study of humoral and T-cell responses to SARS-CoV-2." For statistical reasons, the number of patients should be greater to validate the study.

Authors’ response: We thank the reviewer for this important note. We agree with the reviewer regarding the comment about the small sample size of the subjects. However, due to the very limited availability of imported COVID-19 vaccines in our country, we were unable to add serum samples from such vaccinated individuals to the screening. Therefore, we conducted additional studies of neutralizing activity of sera from individuals vaccinated with Sputnik V vaccine (35 samples) and from individuals who had recovered from COVID-19 caused by SARS-CoV-2 (B.1) (36 patients). New results were also obtained side-by-side using the two test systems and a correlation was found for the calculated IC50 values as well as for the previously screened samples. Given these results, the expanded experimental sample included serum samples from 59 COVID-19 convalescents and 40 vaccinated individuals.

Figure 1 could be improved with particular annotations on Westerblot (figure1C,D,E) for ease of reading.

Authors’ response: We thank the reviewer for noting this. The figure was corrected accordingly. To better illustrate the agreement between the theoretical and observed mass of the streptavidin-eGFP protein, we added SDS-PAGE with a protein molecular weight marker to Figure 1 (panel C). In addition, in Figure 1, D ,E, F, bands corresponding to the active form of EGFP-SA and incorrectly folded aggregates formed during inadequate storage were indicated.

Figure2 : the letters and numbers are not sufficiently legible on the tables (modification of size of letters for example). An explanation on negative and positive controls is to be added (in the text or on figures).

Authors’ response: We thank the reviewer for this critique. Unfortunately, we cannot increase the size of well letters and the numbers indicating the number of FFUs in the resulting images obtained with the vSpot Spectrum Reader. Nevertheless, we added a description of the controls to the corresponding paragraph of the Materials and Methods chapter on performing the microneutralization assay.

Authors’ response: We checked the manuscript for the PLOS ONE's style requirements.

https://pdfs.semanticscholar.org/755b/684fa2e81cfa38957ed82f7fb90ee6d88405.pdf

https://www.mdpi.com/2076-393X/11/12/1810

Authors’ response: We revised our manuscript to reduce similarities with our previous publications.

“This research was funded by the RSCF grant 21-75-30003.”

Authors’ response: We included the following statement to the manuscript: «The funder had no role in study design, data collection and analysis, decision to publish, or preparation of the manuscript.”.

Please confirm at this time whether or not your submission contains all raw data required to replicate the results of your study. Authors must share the “minimal data set” for their submission.

If your submission does not contain these data, please either upload them as Supporting Information files or deposit them to a stable, public repository and provide us with the relevant URLs, DOIs, or accession numbers.

Authors’ response: We changed the Data Availability Statement for «Data are available from corresponding author who may be contacted by email».

Authors’ response: This statement is now only in the Methods section.

6. PLOS ONE now requires that authors provide the original uncropped and unadjusted images underlying all blot or gel results reported in a submission’s figures or Supporting Information files. 

Authors’ response: We included raw blot photos to the submission.

7. Please review your reference list to ensure that it is complete and correct.

Authors’ response: The list of references was revised and corrected.

---

## [Editor Report · Decision Letter 1]

14 May 2024

A novel immunofluorescent test system for SARS-CoV-2 detection in infected cells

PONE-D-24-07586R1

Dear Dr. Rak,

We’re pleased to inform you that your manuscript has been judged scientifically suitable for publication and will be formally accepted for publication once it meets all outstanding technical requirements.

Kind regards,

Boyan Grigorov

Academic Editor

PLOS ONE
---

## [Editor Report · Acceptance letter]

20 May 2024

PONE-D-24-07586R1 

PLOS ONE

Dear Dr. Rak, 

I'm pleased to inform you that your manuscript has been deemed suitable for publication in PLOS ONE. Congratulations! Your manuscript is now being handed over to our production team.

Kind regards, 

on behalf of

Dr. Boyan Grigorov 

Academic Editor

PLOS ONE